# Spine Kinematic Alterations in Nordic Walking Under Two Different Speeds of 3 and 5 km/h—A Pilot Study

**DOI:** 10.3390/jfmk10030330

**Published:** 2025-08-28

**Authors:** Ivan Ivanov, Assen Tchorbadjieff, Oleg Hristov, Petar Peev, Grigor Gutev, Stela Ivanova

**Affiliations:** 1National Sports Academy “Vassil Levski”, 1700 Sofia, Bulgaria; hristov.oleg@gmail.com (O.H.); petar.r.peev@gmail.com (P.P.); grigor.gutev@gmail.com (G.G.); stela_kalinova@abv.bg (S.I.); 2Institute of Mechanics, Bulgarian Academy of Sciences, 1113 Sofia, Bulgaria; 3Institute of Mathematics and Informatics, Bulgarian Academy of Sciences, 1113 Sofia, Bulgaria; atchorbadjieff@math.bas.bg

**Keywords:** nordic walking kinematics, spine movement analysis, upper spine kinematics, lower spine kinematics, pelvis kinematics

## Abstract

Objectives. The present study aimed to quantify changes in spinal kinematics during Nordic walking compared to regular walking (RW) for 60 s on a training path among physically fit young males (n = 20, aged 19–22 years). Methods. Two walking speeds were analyzed: 3 km/h and 5 km/h. The experimental setup was designed to assess spinal angular rotations using five kinematic parameters: upper spine, lower spine, thoracic region, lumbar region, and pelvis. Results. The data were acquired from 9 compact inertial sensors and the following motion analysis is carried out using 3D MioMotion IMU sensor’s analysis system. The differences in the obtained cyclic biomechanical parameters were detected using functional data analysis (FDA) statistical tests. Conclusions. The key finding of the study is that Nordic walking significantly alters the angular kinematic pattern of spinal movement as it revealed significant differences in all five measured parameters when compared to normal walking. Notably, the most pronounced changes were observed in the upper spine and pelvis motion. Additionally, Nordic walking increased stance phase duration and velocity: (i) significantly increased the duration of the stance phase in all three planes of motion; (ii) significantly increased the velocity during the stance phase across all three planes. These reported findings highlight the biomechanical, preventive, therapeutic, and rehabilitative potential of Nordic walking.

## 1. Introduction

Nordic walking (NW) has gained popularity under various names such as ski walking, walking with sticks or fitness walking. This sport appeared with the original purpose of filling the break of the ski season. Nordic walking is a combination of cross-country skiing and regular walking (RW). It can be practiced all year round and anywhere. NW originated in Finland a few decades ago. In recent years, it has become extremely popular and more than 14 million people are already using it as a means of maintaining their condition worldwide. In recent years, NW has been widely used by both athletes in training and by ordinary people for health. It is done with walking sticks, as an aid to traction, better stability and inclusion of the hands when moving [1]. Although relatively recent, its international development is significant due to its advantages [2,3,4,5,6,7].

In recent years, NW has been widely adopted by both professional athletes for training and by the general population for health improvement. The activity involves the use of specially designed poles, which aid in propulsion, enhance stability, and engage the upper limbs during locomotion [1]. Despite being a relatively recent discipline, its international adoption has been significant, largely due to its wide-ranging health benefits [2,3,4,5,6,7,8,9,10,11,12,13,14,15,16,17,18,19,20].

The use of poles in NW provides numerous benefits, including:-Engagement of upper limb muscles [3];-Facilitated movement through reduced muscular effort and lower limb load;-Increased total body energy expenditure due to upper limb involvement;-Reduced mechanical load on lower extremities [4];-Enhanced overall endurance [5];-Decreased vertical ground reaction forces on the legs—NW promotes higher ground reaction force (GRF) and speed compared to RW [6];-Reduced stress on the knee and hip joints [3];-Increased blood lactate concentration during aerobic exercise in individuals with Long-COVID syndrome during this moderate-to-high intensity aerobic exercise therapy program [8,9];-Offer a potentially effective approach for reducing pain and fatigue in individuals living with chronic conditions [10];-Useful therapeutic intervention for improving the exercise capacity and gait technique in male ischemic heart disease patients [11];-The combination of NW training and time-restricted eating is a welltolerated intervention for individuals with an abnormal body composition [12].-Demonstrated notable enhancements in glycemic control and aerobic capacity in individuals with type 2 diabetes and a normal BMI [13];-8.6 months of NW training is effective in preventing/delaying the sarcopenia among postmenopausal women with NAFLD and pre-diabetes [14];-Improved cardiorespiratory endurance, muscle strength, and overall physical performance in both adults and older individuals [15,16];-Enhanced gait and exercise tolerance in patients with cardiovascular diseases [11,17], as well as improved motor function, balance, and quality of life in patients with Parkinson’s disease [18]-Reduced pain and improved physical function in individuals with chronic low back pain and other musculoskeletal disorders [19,20]

Nordic walking (NW) is recognized as a form of physical activity with substantial overall health benefits. It provides superior short- and long-term effects on the cardiorespiratory system compared to RW. Short-term physiological responses include increased heart rate, VO_2_ uptake, respiratory exchange ratio, lactate thresholds, and caloric expenditure, alongside improvements in lipid profiles [7]. Additionally, NW has been shown to enhance core stability [21], positively affecting resting heart rate, blood pressure, exercise capacity, maximal oxygen consumption, and overall quality of life.

NW involves the active engagement of upper body muscles, which remain largely underutilized during RW, while simultaneously reducing the load on the lower extremities [1,2,3]. It recruits over 90% of the body’s muscles—compared to less than 70% during RW. Energy expenditure increases by 20–40%, promoting a balanced full-body workout. NW stimulates the cardiovascular system, improves coordination and balance, strengthens the thighs, knees, and ankles, boosts mood, and helps reduce stress levels. It also facilitates recovery from musculoskeletal pain and lowers the risk of injury [13,22,23,24,25,26].

Various gait modification techniques—such as reduced walking speed, toe-out gait, medial thrust gait, increased trunk sway, and assisted walking with pole supports—have been reported to effectively reduce joint loads and contact forces [27,28,29].

Global literature reports diverse findings regarding the kinematic and biomechanical changes associated with NW in comparison to RW (i.e., without poles) [1,2,3,4,5,6,7,21,22,23,24,25,26,27,28,29,30]. For instance, in a 2024 study involving older adults, Szpala et al. concluded that anticipated biomechanical advantages of NW—such as reduced cadence and enhanced joint mobility—were not observed [30]. However, the study was limited by a small sample size of only seven participants.

From a biomechanical standpoint, NW is frequently compared to RW to assess whether pole use influences joint loading. Most studies focus on the lower limbs, particularly the knee joint, but results remain inconclusive and sometimes contradictory [5,31,32,33,34,35].

Numerous studies have demonstrated the metabolic effects of NW [25,36,37,38,39]. Most report an increase in oxygen consumption of 18–25% compared to walking at the same speed [40,41]. It is also known that energy expenditure in NW varies depending on factors such as terrain inclination [23,42,43]. However, there is a lack of studies exploring whether variations in NW technique affect physiological responses to exercise, although some research suggests this as a plausible hypothesis [25,40].

According to Huang et al., 2021 NW has limited influence on the improvement of spinal posture and back pain in community-dwelling older adults. They conclude that a 12-week Nordic walking training program has a potential to improve upper and lower body strength and balance [20]. Hanuszkiewicz et al., 2020 summarised that NW leads to a statistically significant increase in trunk muscle function compared to a standard general exercise programme but correction of the body posture of women treated for breast cancer was not achieved after the 8-week NW or general exercise interventions [44]. As well Peyré-Tartaruga et al., 2022 found that NW and pole walking using predominantly elbows clearly have larger mediolateral margins of stability than walking, decreasing moderately the muscular engagement of gluteus medius [45].

To date, according to the authors, no data have been found in the literature on changes in five-point spinal kinematics during Nordic walking and comparison with RW. There is a noticeable gap in the literature regarding comparisons between free gait and NW gait in healthy elderly individuals, especially with a focus on spinal kinematics [30]. Most existing studies concentrate on NW as a training modality for improving functional efficiency in elderly populations or for managing conditions associated with circulatory, vascular, musculoskeletal, or neurological disorders [30].

The present study aimed to quantify alterations in spinal kinematics during Nordic walking in comparison to regular walking among physically fit young males, using two walking speeds: 3 km/h and 5 km/h. A functional data analysis approach was employed to analyze the data and identify statistically significant differences in cyclic biomechanical parameters.

## 2. Materials and Methods

### 2.1. Participants and Measurements

In this study were tested twenty (n = 20) male students in the 19–22 age range. They completed an injury record and were informed in detail about the aim of the experiments and the procedure. All participants provided informed consent. Every participant in the target group declared lack of prior experience with Nordic walking and excellent health status. The experimental procedure was approved by the Scientific Council of the NSA “Vassil Levski”, Sofia, Bulgaria.

All experimental measurements were performed on a training path, allowing the selection of a constant walking speed (Figure 1). Two slow walking speeds of 3 and 5 km/h were chosen due to the fact that Nordic walking is practiced unprofessionally mainly by the elderly. The second reason for choosing of this two NW velocities is the fact that they are probably the most used in NW.

The experimental model used contains four types of walking for 60 s, which following one after the other in the following sequence and it was recorded of certain quantitative parameters:-RW at a speed of 3 km/h on a training track for 1 min;-NW at a speed of 3 km/h on a training trail for 1 min;-RW at a speed of 5 km/h on a training track for 1 min;-NW at a speed of 5 km/h on a treadmill for 1 min.

Each of the study participants conducted these four types of walking. An analytical formula was used to determine the length of the poles used, taking into account the height of the particular participant.

Motion analysis is conducted with the 3D MioMotion (MR3.18) IMU sensor analysis system (Noraxon, https://www.noraxon.com/), utilising compact inertial sensors that are positioned on specific body segments as indicated in Figure 2. The obtained kinematic data was acquired with sampling rate 200 Hz. The data was collected from 9 IMU sensors positioned in groups of 3 items on left lower and upper limbs and on the spine (Figure 2). From this interaction structure five parameters are studied in all 3 spatial directions for two different speeds—3 and 5 km/h—Upper spine, Lower Thoracic, Lumbar, Pelvis. All tests were repeated for RW and NW.

### 2.2. Data Models

The standard statistical theory is designed to deal with independent and identically distributed (i.i.d.) data. However, the assumption of independence does not hold for many biomechanical data. As a result, the analysis is performed using Functional data analysis (FDA). It is a branch of statistics which analyzes observations treated as functions, curves, or surfaces. In particular, the time point data obtained in this experiment can be transformed into functional data. This is a usual approach for similar cases. For more information, see [46,47].

Specially, these Nordic walking tests are performed for 20 fit athletic students (n = 20). The measurements are split on two paired experiments for participants walking with speeds of 3 and 5 km/h. They are measured for more than 100 time points. The data is aggregated from all repeated measurements during complete foot step period interval. The final outcome is a periodic time series sampled by more than 100 time points. In aim to get a common grid, the data is aggregated additionally rearranged to 100 equally spaced intervals of relative time [0, 1], representing the complete movement cycle (% stance).

Thus, the used data for every measured parameter consist of l=2 groups and explained by functions $X_{ij}\left(t\right)$, where i=1,…,l and $j=1,\ldots ,n_{l}$. In terms of our paired experiment, l=2, $n_{1}=n_{2}$ and the measurements are bounded in the interval (%stance) $I=\left[0,1\right]$ These functions $X_{ij}\left(t\right)$ are stochastic process, with mean function $\mu_{i}\left(t\right),t\in I$ and covariance function $\gamma \left(s,t\right),s,t\in I$. We follow strictly the definitions and assumptions in [47], where $X_{ij}\left(t\right)$ is defined in $L_{2}\left(I\right)$, a Hilbert space consisting of square integrable functions on I.

There are many methods how to measure the statistical significance for validity of $H_{0}$ against the alternative hypothesis for significant difference ($H_{A}$). All of them rely partly or completely on sample group mean functions:$$\begin{array}{c} \bar{X}\left(t\right)=\frac{1}{n}\sum_{i=1}^{l}\sum_{j=1}^{n_{j}}X_{ij}\left(t\right)\\ {\bar{X}}_{i}\left(t\right)=\frac{1}{n_{i}}\sum_{j=1}^{n_{i}}X_{ij}\left(t\right),i=1,\ldots ,l. \end{array}$$

Between subject ($SSR_{n}\left(t\right)$) and within-subject variations ($SSE_{n}\left(t\right)$) are obtained from:$$SSA_{n}\left(t\right)=\sum_{i=1}^{l}n_{i}{\left({\bar{X}}_{i}\left(t\right)-\bar{X}\left(t\right)\right)}^{2}$$$$SSR_{n}\left(t\right)=\sum_{i=1}^{l}\sum_{j=1}^{n_{i}}{\left(X_{ij}\left(t\right)-{\bar{X}}_{i}\left(t\right)\right)}^{2}$$

The theory has been expanding largely with development of many different tests for years. But most of the popular of them rely on modified pointwise F-test statistics, derived from (1) and (2), and described in [48]. The widely adopted F-type tests are naive (FN) and bias-reduced methods (FB). Another popular approach is to use L2-norm based tests, naïve (L2N) and biased-reduced (*L2B*) variants. All of them are well explained in [48].

The pointwise F-type test tends to be time-consuming, and it does not guarantee that the null hypothesis is significant overall at a given significance level, even if the pointwise F-type test is significant for all t ∈ [0, 1] at the same significance level [49] Due to, the pointwise function has to be integrated in aim to obtain an estimate for overall difference. The basic statistic in this case is obtained from SSA(t) as:$$C_{n}=n\int_{0}^{1}{\left(\bar{X}\left(t\right)-\bar{X}\left(t+1\right)\right)}^{2}dt,t\in \left[0,2\right].$$

However, more powerful criteria is derived from unbiased estimators of covariance function $C\left(s,t\right)$ and asymptotic covariance function $K\left(s,t\right)$:$$\begin{array}{c} C\left(s,t\right)=\frac{1}{n}\sum_{i=1}^{n}\left(\left(X_{i}\left(s\right)-\bar{X}\left(s\right)\right)\left(X_{i}\left(t\right)-\bar{X}\left(t\right)\right)\right),s,t\in \left[0,2\right],\\ K\left(s,t\right)=C\left(s,t\right)-C\left(s,t+1\right)-C\left(s+1,t\right)+C\left(s+1,t+1\right),\;s,t\in \left[0,1\right] \end{array}.$$

Having them, the modified statistics:$$D_{n}\left(t\right)=n\int_{0}^{1}\frac{{\left(\bar{X}\left(t\right)-\bar{X}\left(t+1\right)\right)}^{2}}{K\left(t,t\right)}dt$$

Another, even more powerful statistics can be obtained from the maximum (supremum) of the pointwise F-statistic:$$E_{n}\left(t\right)=Sup_{t\in \left[0,1\right]}\left(\frac{{\left(\bar{X}\left(t\right)-\bar{X}\left(t+1\right)\right)}^{2}}{K\left(t,t\right)}\right)$$

The $E_{n}\left(t\right)$ statistics outperforms $D_{n}\left(t\right)$ in cases of high and moderate correlation between observations at any two different time points. Actually, the tests based on the test statistics $D_{n}$ and $E_{n}$ are more powerful testing procedures than $C_{n}$, which we will not consider. More detailed explanations are available in [50].

For visual data analysis the results are presented with graphs of group means functions and their pointwise SSA(t) statistic plots. Their role is to outline stance (phase) dependent differences between two groups. The L2 norm of the difference between the mean functions of the Normal and Nordic walking groups is reported as part of the graph explanations. The L2-norm mean differences, defined as:$${\left\|\mu_{1}-\mu_{2}\right\|}_{2}=\sqrt{\frac{1}{T}\int_{0}^{T}{\left(\mu_{1}\left(t\right)-\mu_{2}\left(t\right)\right)}^{2}dt.}$$

It quantifies the average Euclidean distance between the mean curves of the Normal and Nordic groups across the 100 time points interval T.

The more powerful estimate for mean differences is obtained from Dn(t) and En(t) tests. They are constructed under the null hypothesis for each parameter at different speed settings:$$H_{0}:\mu_{1}=\mu_{2},t\in I.$$

In this notation, t ∊ [0, 1] corresponds for RW and t ∊ [1,2] for Nordic walking for every test.

Due to small sample size, the computations rely on permutation approach proposed by [50,51]. For this work it is considered P1 permutation model because its best performance for Dn and En in case when l = 2 according [50]. The *p*-values are the proportion of resampled statistics T_b_ that meet or exceed the observed T_obs_:$$p=\frac{1}{B}\sum_{b=1}^{B}I\left(T_{b}\geq T_{obs}\right)$$
over B = 1000 replications. The nominal significance level of 0.05 is chosen. For implementation, it is used the rmfanova library (R package version 0.1.0) [52] in R statistics (version 4.4.3) [53].

Worth noting, that the obtained (reported) results, presented as SSA(t) plots and *p*-values, provide estimates of group differences. However, a closer examination of individual data records reveals unique patterns and individual differences that are not addressed in this study.

## 3. Results

The experimental setup employed in this study was specifically designed to evaluate spinal angular rotations across all three planes (Figure 3). By comparing the results obtained for the two types of walking, we obtained information about the 3D Spine kinematic alterations in NW under two different speeds of 3 and 5 km/h for five parameters studied Upper spine, Lower spine, Thoracic, Lumbar and Pelvis.

### 3.1. Upper Spine

By examining the upper spine graphs obtained (Figure 4, Figure 5 and Figure 6) we can note especially in the case of upper spine pitch (Figure 5) that we have a real difference of around 3 degrees angle for whole cycle which should be put on quantitatively.

After applying the modified statistical approach the obtained data for all 3 directions are plotted for all measured trajectories and mean values for RW and NW samples (Figure 4, Figure 5 and Figure 6).

All statistics show statistical differences (Table 1) at particular parts of rotation cycle detected by SSA(t)—degrees of squared error of the angle difference (Y axes scale of SSA(t)). The all 3-direction plots and $SSA\left(t\right)$ results are listed below in following order—course, pitch, roll for both walking speeds.

The pointwise statistic’s estimates in which part of the cycle we have clearly expressed statistically significant differences in SSA(t). It can be seen that this statistical differences varies according the concrete cycle. The level of the maximum of the curves on Figure 7, Figure 8 and Figure 9 (as with all similar graphs further down in the text for the other measured parameters) indicates a significant difference at the corresponding location of the corresponding cycle for concrete parameter. The maximum values of the SSA(t) curve show the cycle points at which we have the maximum statistically significant difference between the compared walking modes. The minimum values of the curve (near to 0) show the cycle points at which the twist trajectories coincide.

For instance, for Upper spine pitch we can see that throughout the entire cycle we have a statistically significant difference in favor of 20–50 degrees of squared error of the angle difference (Y axes scale of SSA(t)). This is not the case with the other two parameters. This means that the statistical model proposed by us sensitively and accurately reflects the differences throughout the entire walking cycle.

The structure of the results presentation for the remaining four parameters is the same as the presentation in this paragraph.

### 3.2. Lower Spine

The plots for lower spine course, pitch and roll data of two walking velocities (3 km/h data is on left, 5 km/h data is on right) are shown in Figure 10, Figure 11 and Figure 12.

There are some particular points on cycle range that imply differences. They are outlined by the highest values of pointwise SSA(t) plots (Figure 13, Figure 14 and Figure 15).

### 3.3. Thoracic

The plots for thoracic flexion, axial and lateral data of two walking velocities (3 km/h data is on left, 5 km/h data is on right) are shown in (Figure 16, Figure 17 and Figure 18).

There are some particular points on stance cycle range that imply differences. They are outlined by the highest values of pointwise SSA(t) and F(t) plots (Figure 19, Figure 20 and Figure 21).

The all 3-direction plots and results are listed bellow in following order—flexion, axial, lateral rotations for both walking speeds.

### 3.4. Lumbar

The plots for lumbar flexion data of two walking velocities (3 km/h data is on left, 5 km/h data is on right) (Figure 22, Figure 23 and Figure 24) is below with respective SSA(t) statistics (Figure 25, Figure 26 and Figure 27):

### 3.5. Pelvis

The pelvis tests result for all 3 directions are listed below (Figure 28, Figure 29 and Figure 30) is below with respective SSA(t) statistics (Figure 31, Figure 32 and Figure 33). The plots for pelvis course data of two walking velocities (3 km/h data is on left, 5 km/h data is on right) is below:

All pointwise SSA(t) statistics show differences at particular parts of rotation cycle (detected as higher values) (Table 1). The all 3-direction plots and results are listed below in following order—course, pitch, roll for both walking speeds.

The outlined differences in plotted pointwise SSA(t) results imply that walking with poles significantly changes the kinematic structure of walking. This observation is tested for all studied parameters and confirmed numerically for many of them. The tests’ results are presented by their *p*-values obtained from D_n_(t) and E_n_(t) statistics in Table 1.

These changes, as we showed above, have an angular component, but what happens in the relationship between the two phases of walking, the stance phase and the swing phase, is interesting to be clarified. So we included in our study an additional Mann-Whitney U statistical comparison between RW and NW for three parameters—Range, Duration and Velocity of the walk cycle for upper spine course parameter. These three parameters were obtained for every tested participant from Noraxon software automatically.

The additional comparison is performed using Mann-Whitney U test due to small sample size. The tested Null hypothesis (H_0_) is that both paired groups come from the same non-parametric distribution with no difference in medians. All results are tabulated in Table 2.

The results obtained in this statistical comparison (Table 2) strictly confirmed the change in the kinematic temporal structure of walking in both the stance phase and the swing phase.

## 4. Discussion

The pointwise SSA(t) graphs illustrate the specific phases of the gait cycle where statistically significant differences occur, complementing the overall results presented in Table 1. For instance, the peaks in Figure 7, Figure 8 and Figure 9 (as well as the similar trends in Figure 13, Figure 14, Figure 15, Figure 19, Figure 20, Figure 21, Figure 25, Figure 26, Figure 27, Figure 31, Figure 32, and Figure 33) indicate the locations within the cycle where statistically significant differences are observed for specific parameters. This provides more detailed insights into exactly where in the cycle these differences occur, enhancing our understanding of how NW affects kinematic patterns in tested participants.

From a practical perspective, these results highlight the phases of the gait cycle where NW exerts a pronounced influence on spinal movement and kinematic structure compared to RW.

According to Huang et al. (2021) NW requires a forward torso inclination, which typically results in a reduced torso flexion angle [20,54] and increased activation of trunk muscles [44]. However, scientific studies examining the effect of NW on spinal posture remain limited [21,44,55,56,57]. Furthermore, the observed effect of NW has been reported on groups of elderly people with pathological conditions with different individual level, after NW program for a period of time [44,54,57]. The findings of Huang et al. (2021) [20], which reported no significant changes in spinal posture, align with previous studies showing that 6–8 weeks of NW training did not significantly alter spinal posture in healthy older adults [56] or in older women with Parkinson’s disease [57]. In contrast, our study covers a target group of completely healthy young students with a good fitness status and similar age. Another significant difference of our experimental approach is that the two compared modes—RW and NW follow one after the other. Such a sequence has the advantage of allowing the detection of kinematic changes that occurred solely as a result of the use of poles. In this way, two undesirable factors are avoided. The first is the existing pathology and its varying degree in each individual participant in the cited articles [21,44,57]. The second is the strictly individual degree of adaptation over time to poles of each NW practitioner. These important considerations, reflected in our experimental approach, allow us to synthesize conclusions regarding changes in both the angular spinal kinematics (Table 1) and the temporal structure of pole walking (Table 2).

Our results clearly demonstrate that the kinematic structure of walking is significantly altered when comparing NW to RW at both speeds of 3 km/h and 5 km/h. The chosen IMU sensor configuration allowed us to collect precise data from five critical anatomical locations that characterize the walking kinematics. The statistical estimates are obtained from computed *p*-values of Dn(t) and En(t) as they are defined in (2) and (3) (see Table 1).

In an attempt to summarize the obtained tabular results, it can be said that walking with poles significantly changes the kinematic structure of walking in healthy persons—an important finding that supports the health and functional benefits of NW.

These changes have an angular kinematic component. The additional statistical significances at comparison between RW and NW for three parameters—Range, Duration and Velocity of the walk cycle for upper spine course confirm changed angular spine kinematics, showing temporal component. The results obtained in this statistical comparison strictly confirmed the change in the kinematic structure of walking in both the stance phase and the swing phase (Table 2).

Based on the obtained results, important and detailed conclusions can be drawn regarding changes in spinal angular kinematics during Nordic Walking (NW) using poles. Angular changes were observed across five measured parameters (Table 1):

Upper Spine. Statistically significant differences were observed in all three rotational directions (pitch, roll, and yaw) at both walking speeds.

Axial Rotation (Course)—NW results in reduced body rotation in the horizontal plane at both walking speeds (Figure 4). The most statistically significant differences occur at 70% of the gait cycle at 3 km/h and at 75% at 5 km/h (Figure 7).

Pitch (Flexion/Extension)—NW leads to an increase in sagittal plane rotation while maintaining the shape of the angular displacement profile (Figure 5). The maximum statistically significant differences are at 45% (3 km/h) and 50% (5 km/h) of the gait cycle (Figure 8).

Lower Spine. Significant differences were also found in all three directions, except for the pitch direction (DN.p value = 0.107) at 3 km/h.

Axial Rotation (Course)—NW reduces body rotation in the horizontal plane at 3 km/h (Figure 10). The most statistically significant differences occur at 30% and 80% of the gait cycle (Figure 13).

Pitch (Flexion/Extension)—NW increases sagittal plane rotation while preserving the profile of movement (Figure 11). Maximum significant differences appear at the beginning and end of the gait cycle at both speeds (Figure 14).

Roll (Lateral Flexion)—NW reduces frontal plane rotation at both speeds (Figure 12). The most significant differences are at 25% (3 km/h) and at 25% and 60% (5 km/h) of the gait cycle (Figure 15).

Thoracic Region. Statistically significant differences were observed in all directions except for DN.p and EN.p values in the axial (3 km/h) and lateral (3 km/h) rotations.

Flexion—NW increases flexion angle while maintaining the shape of the curve throughout the cycle (Figure 16). The most statistically significant differences are observed at 45% of the gait cycle at both speeds (Figure 19).

Lumbar Region. Significant differences were found in all directions for DN.p and EN.p values.

Flexion—NW reduces lumbar flexion at 3 km/h (Figure 22). The most significant differences are observed at 60% of the gait cycle at 3 km/h and at 15% and 30% at 5 km/h (Figure 25).

Axial Rotation—NW decreases lumbar axial rotation at both speeds (Figure 23). The most significant differences occur at 30% of the cycle at 3 km/h and 20% at 5 km/h (Figure 26).

Pelvis. Here, statistically significant differences were observed in all three compared rotational directions.

Axial Rotation (Course)—NW decreases body rotation in the horizontal plane at both speeds (Figure 28). Significant differences are observed at 75% (3 km/h) and 60% (5 km/h) of the gait cycle (Figure 31).

Pitch—NW reduces sagittal plane rotation at both speeds (Figure 29). The most significant differences occur at 50% (3 km/h) and 45% (5 km/h) of the gait cycle (Figure 32).

Roll—NW increases frontal plane rotation while maintaining the profile of motion (Figure 30). Significant differences are seen at 30% of the gait cycle at both speeds (Figure 33).

### 4.1. Limitations of the Study

The limitations of this research are in four directions. The first is related to the used experimental model, namely the use of a treadmill for the research, which introduces uncertainty. However, in our opinion, this uncertainty does not change the significance of the data obtained and does not invalidate the truth of the conclusions drawn. The second important limitation is the target group and its size. It consists of only fit young men. Considering female and older in age participants may deliver more interesting observations and conclusions. Also, the number of participants is limited to 20. Despite of applied permutations for computations of D_n_(t) and E_n_(t) statistics, it is only partially substitutes the advantage of higher sample volumes. The third is the brief duration of the experiment, which may not adequately capture long-term biomechanical adaptations. The last is the lack of data on upper limb mechanics, an essential component of Nordic Walking.

### 4.2. Hypothesis for Different Kinematic Response Under NW of Different Tested Segments of the Spine Under NW

The results obtained in this work raise more questions than they provide answers. The author team refrained from hypotheses due to the described limitations of the study, due to the specificity of the experiments performed with the correspondence of the small group we included in the research and due to the fact that it is very easy to make hypotheses but difficult to prove them. However, a more general hypothesis can be defined.

The different kinematic response of different tested segments of the spine (e.g., thoracic vs. lumbar) is most likely due to:-the different correlation of these spine segments to the upper limbs and lower limbs movement;-the height and weight of the person and the length of the poles, respectively;-the health status of the person performing NW;-the speed of NW and the terrain on which it is performed;-the strictly individual gait of each person.

## 5. Conclusions

The most important conclusion of the study is that Nordic Walking significantly changes the kinematic structure of spinal movement. The proposed statistical approach confirmed statistically significant differences in all five measured parameters compared to RW. Notably, the most substantial differences were observed in the upper spine and pelvis. The NW modified spinal kinematics at both speeds tested adaptations are highly individual, depending on each person’s physical condition and personal pole usage technique. In some cases, these biomechanical changes are extremely pronounced, highlighting the individualized nature of NW’s impact.

Temporal changes were observed across three measured step phases parameters (Table 2):-Significantly decrease the range of the stance and swing phases in the frontal plane.-Significantly increase the stance phase duration in all three planes of motion.-Significantly decrease the swing phase duration in all three planes.-Significantly increase stance phase velocity in all three planes.-Alter step kinematics, depending on walking speed.

Spine kinematics is crucial in maintaining both functional mobility and general health. Alterations in spine kinematics could impact back pain, decreased mobility, and risk of falls. Nordic walking presents potential advantages for improving gait mechanics and spinal stability. These findings emphasize the biomechanical, preventive, therapeutic, and rehabilitative potential of NW.

How can NW be useful in the rehabilitation of particular conditions such as older adults, neurological diseases, and subjects with a higher fall risk? NW will certainly have a positive effect on pathological conditions of the spine, neurological diseases, but not in the acute phase and tailored to the individual characteristics of the person. Its appointment by clinicists and its dosage should be carried out taking into account the respective pathology and its degree, patient feedback and the degree of the achieved result.

### Future Research Directions

The current measurements and conclusions suggest that future research should aim to further clarify the trends identified in this study. Incorporating additional biomechanical analysis methods will be essential for gaining a more comprehensive understanding of spinal and whole-body biomechanics during Nordic Walking.

## Figures and Tables

**Figure 1 jfmk-10-00330-f001:**
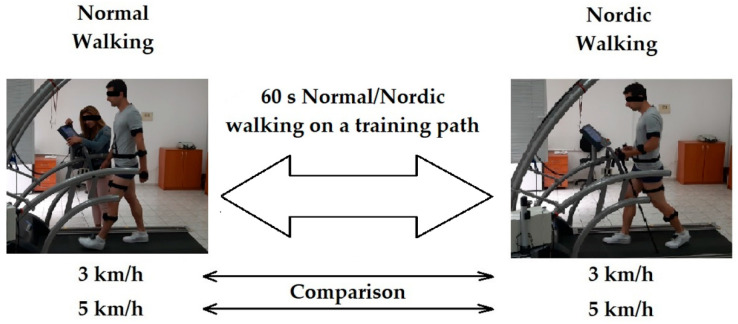
Experimental structure of data collection.

**Figure 2 jfmk-10-00330-f002:**
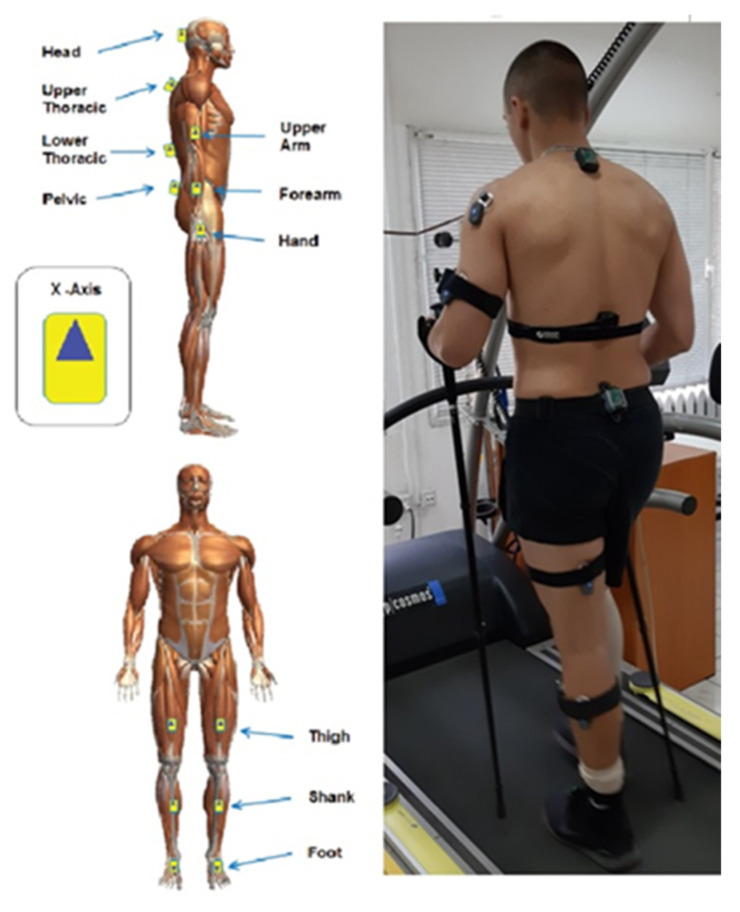
IMU senzors placement geometry.

**Figure 3 jfmk-10-00330-f003:**
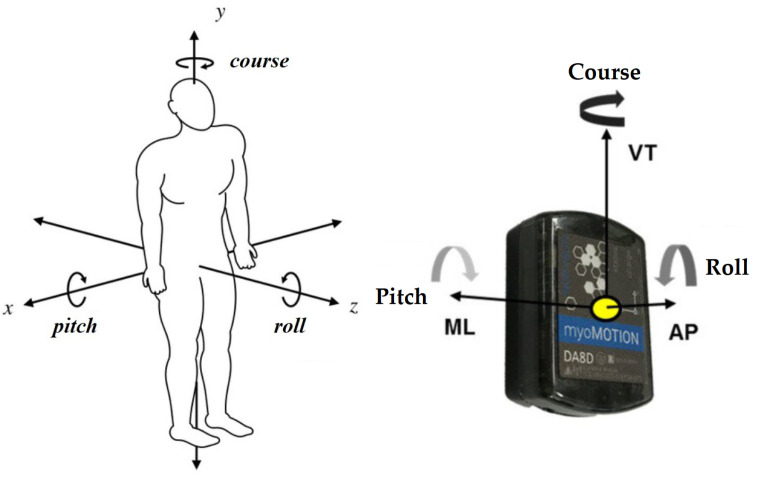
IMU senzor movement axis for data asquisition [54,55].

**Figure 4 jfmk-10-00330-f004:**
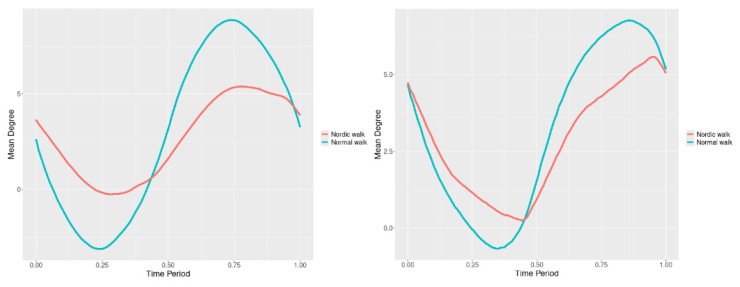
Upper spine course mean functions of two walking velocities (3 km/h data is on (**left**), 5 km/h data is on (**right**). The L2-norm mean differences are 2.546 for the 3 km/h speed test and 1.237 for the 5 km/h speed test.

**Figure 5 jfmk-10-00330-f005:**
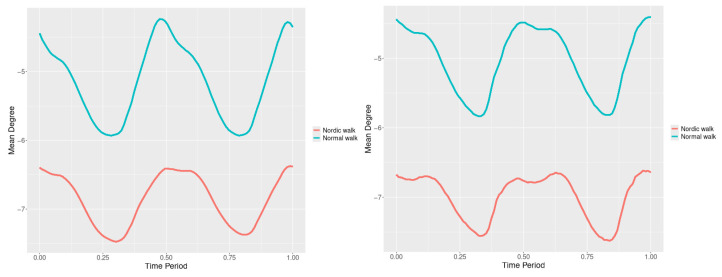
Upper spine pitch mean functions of both walking velocities (3 km/h data is on (**left**), 5 km/h data is on (**right**)). The L2-norm mean differences are 1.733 for the 3 km/h speed test and 1.97 for the 5 km/h speed test.

**Figure 6 jfmk-10-00330-f006:**
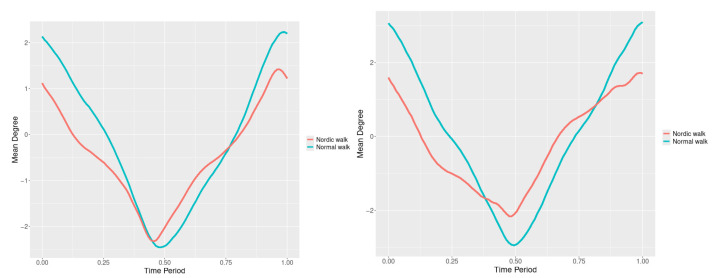
Upper spine roll mean functions of two walking velocities (3 km/h data is on (**left**), 5 km/h data is on (**right**)). The L2-norm mean differences are 0.649 for the 3 km/h speed test and 0.95 for the 5 km/h speed test.

**Figure 7 jfmk-10-00330-f007:**
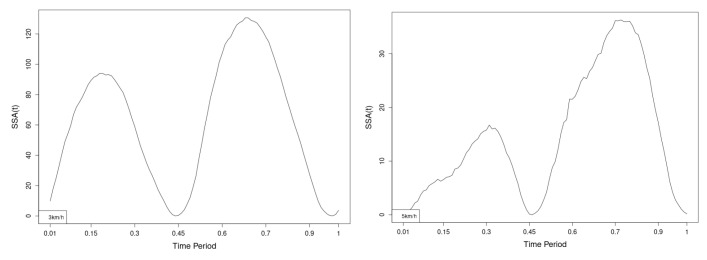
Pointwise SSA(t) statistics for upper spine course angle at 3 and 5 km/h.

**Figure 8 jfmk-10-00330-f008:**
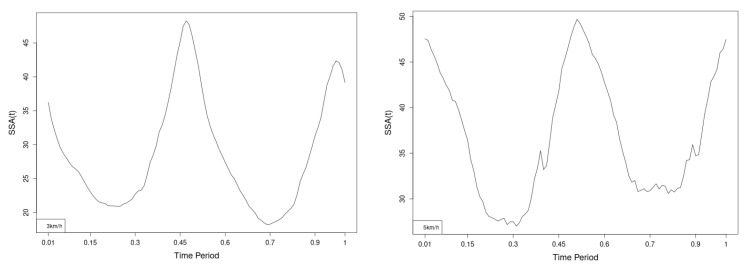
Pointwise SSA(t) statistics for upper spine pitch angle at 3 and 5 km/h.

**Figure 9 jfmk-10-00330-f009:**
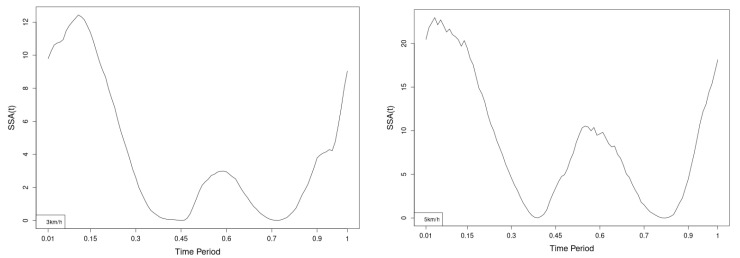
Pointwise SSA(t) statistics for upper spine roll angle at 3 and 5 km/h.

**Figure 10 jfmk-10-00330-f010:**
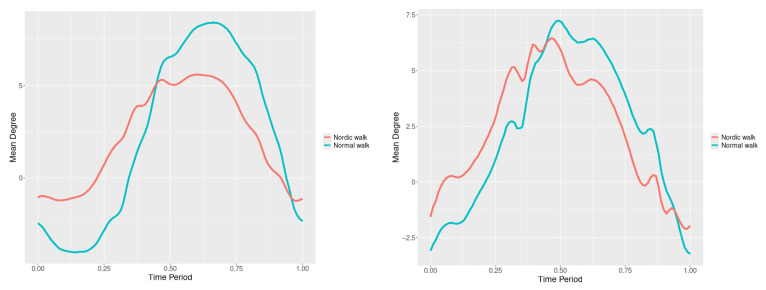
Lower spine course mean functions of two walking velocities (3 km/h data is on (**left**), 5 km/h data is on (**right**)). The L2-norm mean differences are 2.64 for the 3 km/h speed test and 1.731 for the 5 km/h speed test.

**Figure 11 jfmk-10-00330-f011:**
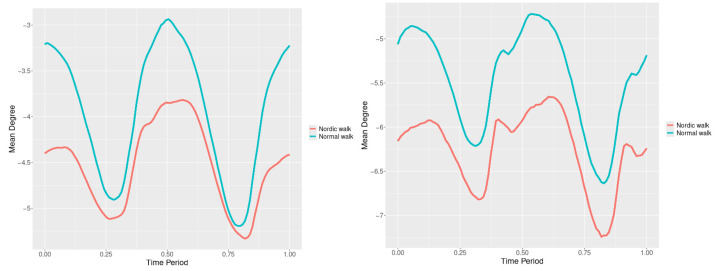
Lower spine pitch mean functions of two walking velocities (3 km/h data is on (**left**), 5 km/h data is on (**right**)). The L2-norm mean differences are 0.693 for the 3 km/h speed test and 0.825 for the 5 km/h speed test.

**Figure 12 jfmk-10-00330-f012:**
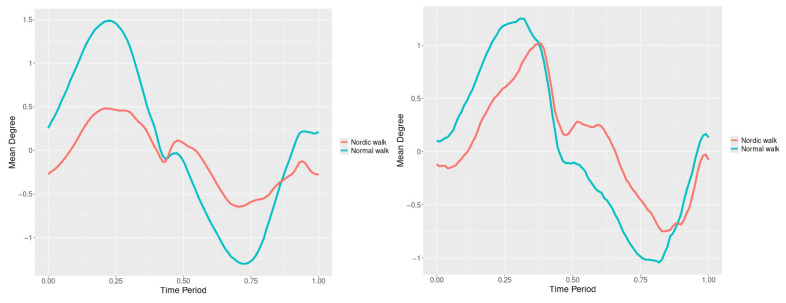
Lower spine roll mean functions of two walking velocities (3 km/h data is on (**left**), 5 km/h data is on (**right**)). The L2-norm mean differences are 0.61 for the 3 km/h speed test and 0.416 for the 5 km/h speed test.

**Figure 13 jfmk-10-00330-f013:**
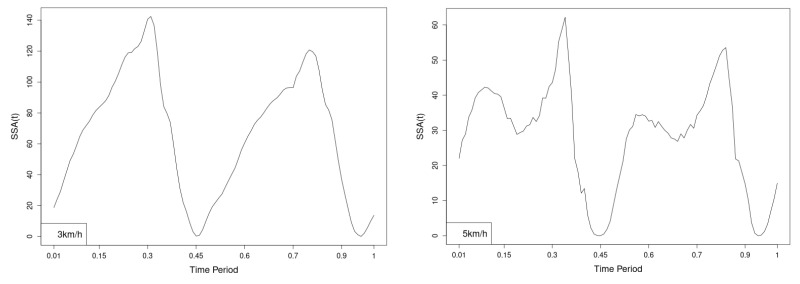
Pointwise SSA(t) statistics for lower spine course angle at 3 and 5 km/h.

**Figure 14 jfmk-10-00330-f014:**
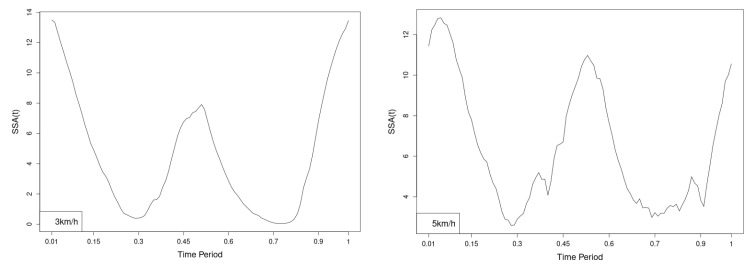
Pointwise SSA(t) statistics for lower spine pitch angle at 3 and 5 km/h.

**Figure 15 jfmk-10-00330-f015:**
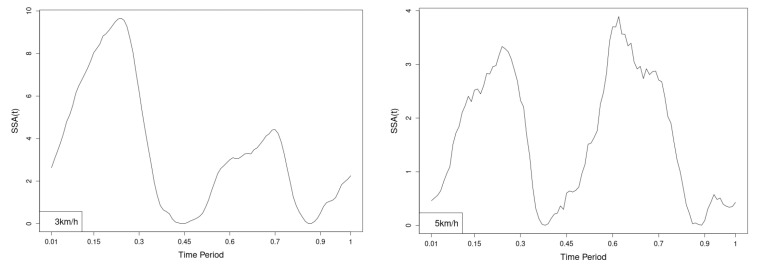
Pointwise SSA(t) statistics for lower spine roll angle at 3 and 5 km/h.

**Figure 16 jfmk-10-00330-f016:**
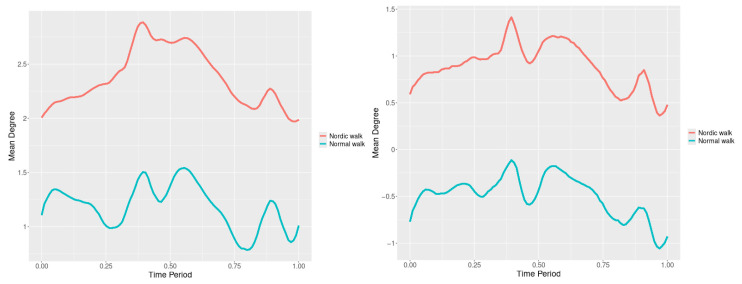
The plots for thoracic flexion mean functions of two walking velocities (3 km/h data is on (**left**), 5 km/h data is on (**right**)). The L2-norm mean differences are 1.203 for the 3 km/h speed test and 1.39 for the 5 km/h speed test.

**Figure 17 jfmk-10-00330-f017:**
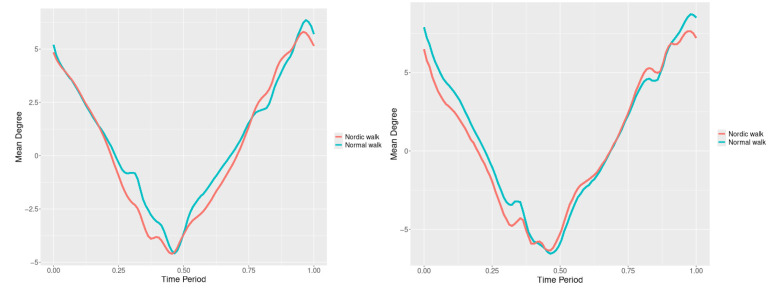
The plots of thoracic axial mean functions of two walking velocities (3 km/h data is on (**left**), 5 km/h data is on (**right**)). The L2-norm mean differences are 0.612 for the 3 km/h speed test and 0.815 for the 5 km/h speed test.

**Figure 18 jfmk-10-00330-f018:**
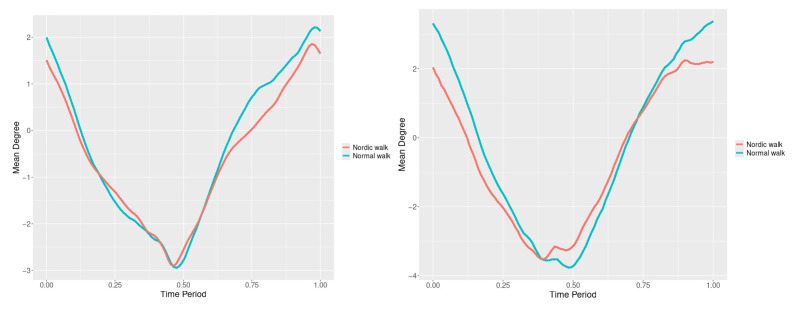
The plots of thoracic lateral mean functions of two walking velocities (3 km/h data is on (**left**), 5 km/h data is on (**right**)). The L2-norm mean differences are 0.329 for the 3 km/h speed test and 0.665 for the 5 km/h speed test.

**Figure 19 jfmk-10-00330-f019:**
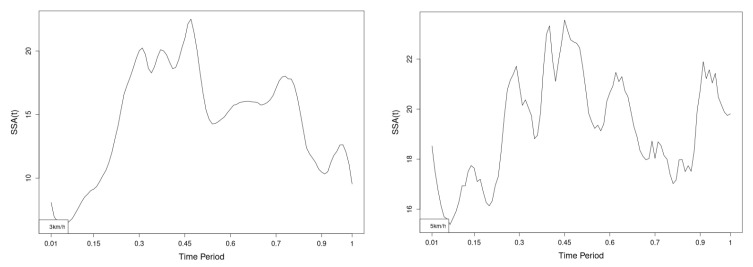
Pointwise SSA(t) statistics for flexion rotation at 3 and 5 km/h.

**Figure 20 jfmk-10-00330-f020:**
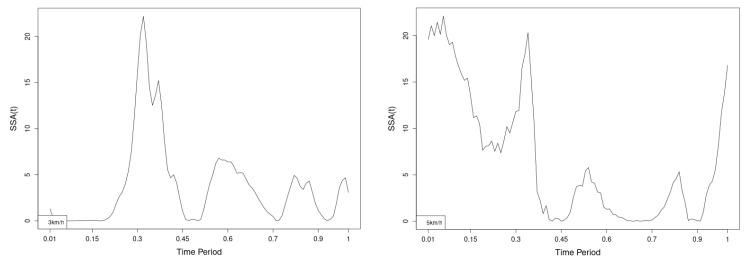
Pointwise SSA(t) statistics for axial rotation at 3 and 5 km/h.

**Figure 21 jfmk-10-00330-f021:**
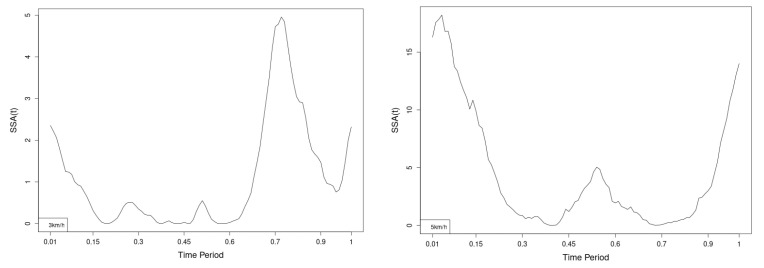
Pointwise SSA(t) statistics for lateral rotation at 3 and 5 km/h.

**Figure 22 jfmk-10-00330-f022:**
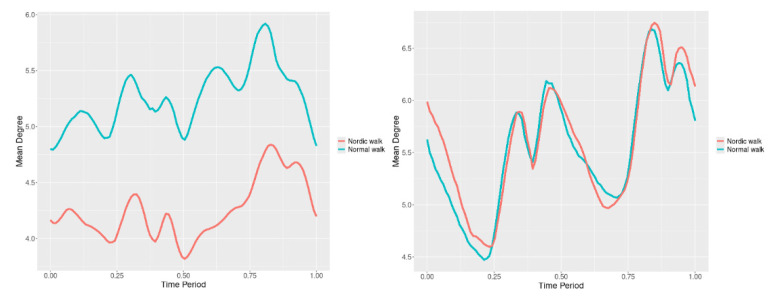
The plots for lumbar flexion mean functions of two walking velocities (3 km/h data is on (**left**), 5 km/h data is on (**right**)). The L2-norm mean differences are 1.026 for the 3 km/h speed test and 0.181 for the 5 km/h speed test.

**Figure 23 jfmk-10-00330-f023:**
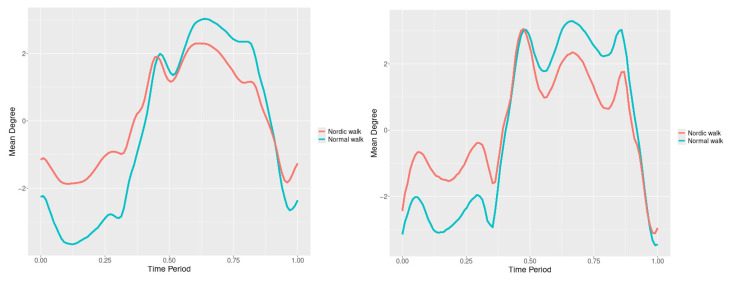
The plots of lumbar axial mean functions of two walking velocities (3 km/h data is on (**left**), 5 km/h data is on (**right**)). The L2-norm mean differences are 1.168 for the 3 km/h speed test and 1.141 for the 5 km/h speed test.

**Figure 24 jfmk-10-00330-f024:**
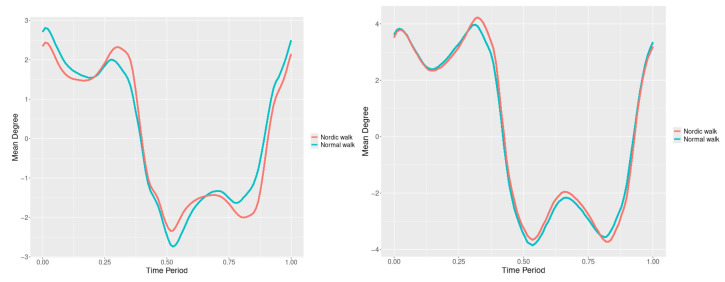
The plots of lumbar lateral mean functions of two walking velocities (3 km/h data is on (**left**), 5 km/h data is on (**right**)). The L2-norm mean differences are 0.363 for the 3 km/h speed test and 0.238 for the 5 km/h speed test.

**Figure 25 jfmk-10-00330-f025:**
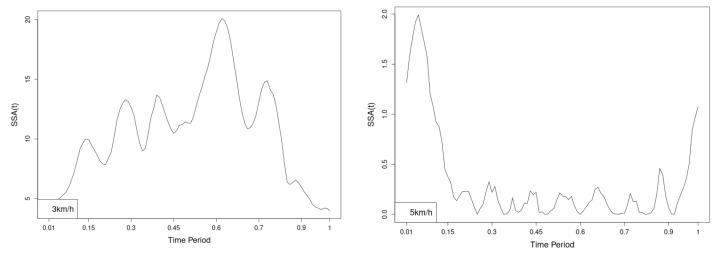
Pointwise SSA(t) statistics for lumbar flexion rotation at 3 and 5 km/h.

**Figure 26 jfmk-10-00330-f026:**
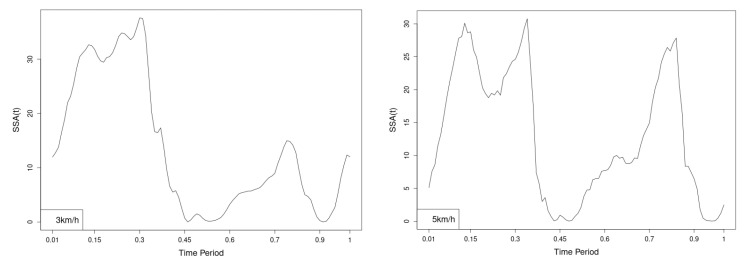
Pointwise SSA(t) statistics for lumbar axial rotation at 3 and 5 km/h.

**Figure 27 jfmk-10-00330-f027:**
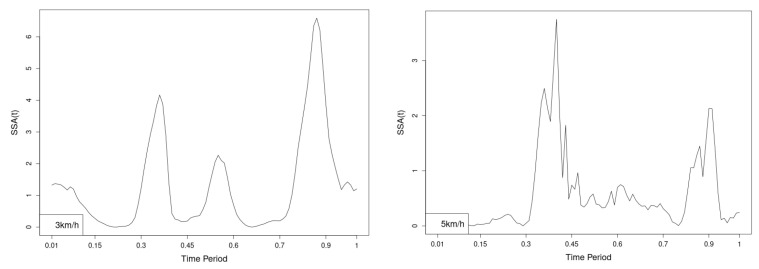
Pointwise SSA(t) statistics for lumbar lateral rotation at 3 and 5 km/h.

**Figure 28 jfmk-10-00330-f028:**
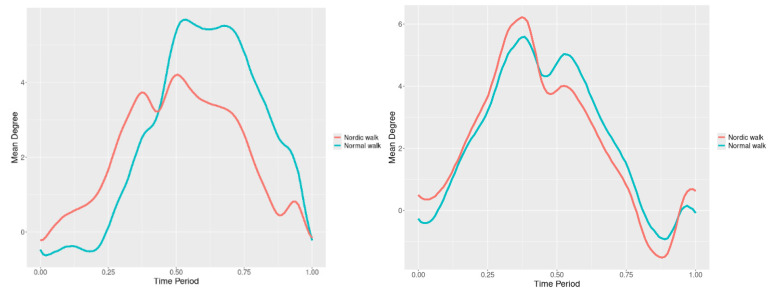
The plots of pelvis course mean functions of two walking velocities (3 km/h data is on (**left**), 5 km/h data is on (**right**)). The L2-norm mean differences are 1.556 for the 3 km/h speed test and 0.646 for the 5 km/h speed test.

**Figure 29 jfmk-10-00330-f029:**
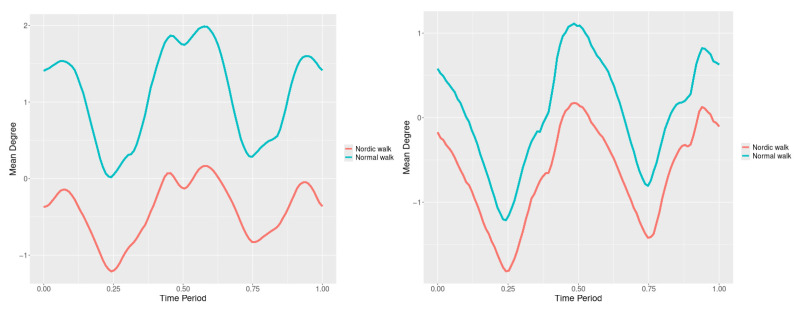
The plots of pelvis pitch mean functions of two walking velocities (3 km/h data is on (**left**), 5 km/h data is on (**right**)). The L2-norm mean differences are 1.537 for the 3 km/h speed test and 0.745 for the 5 km/h speed test.

**Figure 30 jfmk-10-00330-f030:**
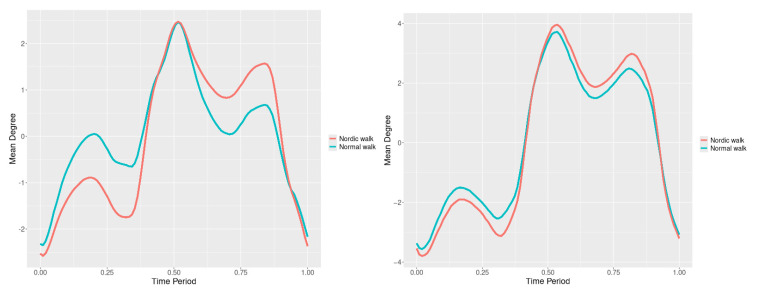
The plots of pelvis roll mean functions of two walking velocities (3 km/h data is on (**left**), 5 km/h data is on (**right**)). The L2-norm mean differences are 0.68 for the 3 km/h speed test and 0.373 for the 5 km/h speed test.

**Figure 31 jfmk-10-00330-f031:**
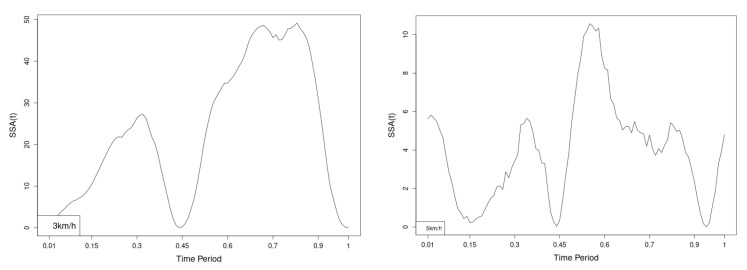
Pointwise SSA(t) statistics for pelvis course angle at 3 and 5 km/h.

**Figure 32 jfmk-10-00330-f032:**
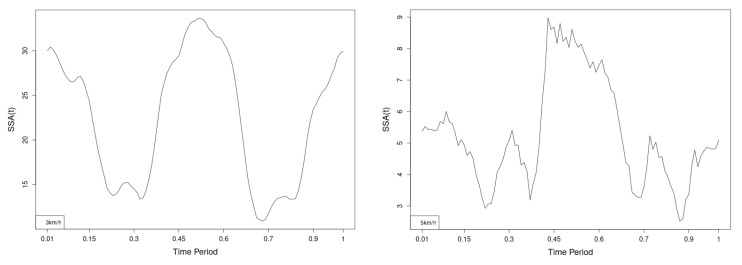
Pointwise SSA(t) statistics for pelvis pitch angle at 3 and 5 km/h.

**Figure 33 jfmk-10-00330-f033:**
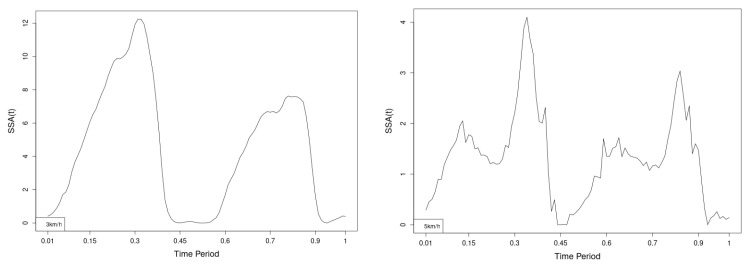
Pointwise SSA(t) statistics for pelvis roll angle at 3 and 5 km/h.

**Table 1 jfmk-10-00330-t001:** *p*-values of five observed parameters in three directions under D_n_(t) and E_n_(t) tests. The stars’ notation is used to mark the significance (* *p* ≤ 0.05 and ** *p* ≤ 0.01).

	Normal 3 km/h—Nordic 3 km/h		Normal 5 km/h—Nordic 5 km/h	
Statistic Protocol	D_n_(t) (*p*-Value)	E_n_(t) (*p*-Value)	D_n_(t) (*p*-Value)	E_n_(t) (*p*-Value)
Upper spine course	0.000 **	0.000 **	0.000 **	0.000 **
Upper spine pitch	0.001 **	0.000 **	0.000 **	0.001 **
Upper spine roll	0.01 **	0.007 *	0.000 **	0.000 **
Lower spine course	0.000 **	0.000 **	0.000 **	0.000 **
Lower spine pitch	0.107	0.047 *	0.020 *	0.010 **
Lower spine roll	0.000 **	0.000 **	0.006 **	0.026 *
Thoracic axial	0.209	0.129	0.009 **	0.011 *
Thoracic flexion	0.011 *	0.008 *	0.004 **	0.006 *
Thoracic lateral	0.188	0.073	0.004 **	0.000 **
Lumbar axial	0.000 **	0.001 **	0.000 **	0.000 **
Lumbar flexion	0.033 *	0.030 *	0.707	0.567
Lumbar lateral	0.006 **	0.002 **	0.161	0.068
Pelvis course	0.000 **	0.000 **	0.006 **	0.016 *
Pelvis pitch	0.000 **	0.000 **	0.000 **	0.000 **
Pelvis roll	0.000 **	0.000 **	0.000 **	0.000 **

**Table 2 jfmk-10-00330-t002:** Measured walking upper spine characteristics—range, duration and velocity in comparison between RW and NW for stance and swing walking phases. The plus sign indicates that the corresponding parameter is increased after NW. The minus sign indicates that the corresponding parameter is decreased after NW. The star’s notation is used to mark the significance (* *p* ≤ 0.05 and ** *p* ≤ 0.01).

Upper Spine Course 3 km/h	Upper Spine Course 5 km/h	Upper Spine Pitch 3 km/h	Upper Spine Pitch 5 km/h	Upper Spine Roll 3 km/h	Upper Spine Roll 5 km/h
Stance phase range 0.0068 * −	Stance phase range 0.0762 −	Stance phase range 0.2627 −	Stance phase range 0.4222 −	Stance phase range 0.0419 * −	Stance phase range 0.0095 * −
Swing phase range 0.0002 ** −	Swing phase range 0.0569 −	Swing phase range 0.2111 −	Swing phase range 0.695 −	Swing phase range 0.0027 ** −	Swing phase range 0.0012 ** −
Stance phase Duration 0.0062 * +	Stance phase Duration 0.0061 * +	Stance phase Duration 0.0055 * +	Stance phase Duration 0.0061 * +	Stance phase 0.0062 * +	Stance phase 0.0061 * +
Swing phase Duration 0.0079 * −	Swing phase Duration 0.0045 ** −	Swing phase Duration 0.007 * −	Swing phase Duration 0.0045 ** −	Swing phase Duration 0.0079 * −	Swing phase Duration 0.0045 ** −
Stance phase Velocity 0.0196 * +	Stance phase Velocity 0.0594 +	Stance phase Velocity 0.0217 * +	Stance phase Velocity 0.013 * +	Stance phase Velocity 0.0196 * +	Stance phase Velocity 0.0054 * +
Swing phase Velocity 0.0145 * −	Swing phase Velocity 0.0594 −	Swing phase Velocity 0.1044	Swing phase Velocity 0.0239 *	Swing phase Velocity 0.0239 *	Swing phase Velocity 0.0095 * −

## Data Availability

The original contributions presented in this study are included in the article. Further inquiries can be directed to the corresponding author.

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
