# Peer review of "Spine Kinematic Alterations in Nordic Walking Under Two Different Speeds of 3 and 5 km/h—A Pilot Study"

_jfmk, 2025, doi:10.3390/jfmk10030330_

Round 1

Reviewer 1 Report

Comments and Suggestions for Authors

This study investigates how walking speed affects spine kinematics in Nordic Walking (NW), comparing two speeds (3 and 5 km/h). This is a relevant and novel topic due to the growing popularity of NW and its use in rehabilitation and fitness contexts. However, the manuscript suffers from methodological weaknesses, superficial analysis, and inadequate discussion of practical implications.

Introduction:

  1. The introduction is too long and hard to read. Also, it is too general at the beginning. Authors should focus more on the relevant studies, rather than explain some basic, already existing knowledge about the topic.
  2. This manuscript lacks a focused rationale for spine-specific analysis. For example, why spine and not lower limb joints, which are more studied? Please include a more precise justification for examining spine kinematics specifically. Also, elaborate on why the chosen speeds (3 and 5 km/h) were selected.
  3. Please add significance to the study after the aims, not the methods or statistical analyses. Also, suggest some hypotheses based on the previous research.

Methods:

  1. Sample size was not justified. Please provide an adequate power analysis for this sample.
  2. There are no participant characteristics beyond age and gender: training history? NW experience? Health status?
  3. It remains unclear if randomisation or familiarisation was used for walking speed order.
  4. There is a lack of clarity on data processing: e.g., how was kinematic data filtered? What was the sampling rate?
  5. The statistical approach is under-described. Repeated-measures ANOVA is mentioned, but assumptions (normality, sphericity) are not addressed, and effect sizes are not reported. On the other hand, I find it unnecessary to add formulas for ANOVA and the T-test. Please rewrite this section to focus more on adequate analyses, post hoc, and reports.

Results:

  1. Figures and tables are dense and not reader-friendly. Results are only briefly described in text, relying on the reader to interpret complex tables and graphs. Please add some summary information in the text or condense the results.

Discussion and Conclusion:

  1. The discussion overgeneralizes from minimal or non-significant findings.
  2. No discussion of why specific spine segments (e.g., thoracic vs. lumbar) respond differently.
  3. After the correction in the statistical analysis, in particular, adding the effect size and justifying the ANOVA in the first place, the discussion should be thoroughly checked and rewritten.

Overall, my recommendation is a major revision.

Author Response

Reviewer 1

This study investigates how walking speed affects spine kinematics in Nordic Walking (NW), comparing two speeds (3 and 5 km/h). This is a relevant and novel topic due to the growing popularity of NW and its use in rehabilitation and fitness contexts. However, the manuscript suffers from methodological weaknesses, superficial analysis, and inadequate discussion of practical implications.

Introduction:

The introduction is too long and hard to read. Also, it is too general at the beginning. Authors should focus more on the relevant studies, rather than explain some basic, already existing knowledge about the topic.

We thank the reviewer, in view of his correction, the text was added:

According to Huang et al., 2021 NW has limited influence on the improvement of spinal posture and back pain in community-dwelling older adults. They conclude that a 12-week Nordic walking training program has a potential to improve upper and lower body strength and balance [20].  Hanuszkiewicz et al., 2020 summarised that NW leads to a statistically significant increase in trunk muscle function compared to a standard general exercise programme but correction of the body posture of women treated for breast cancer was not achieved after the 8-week NW or general exercise interventions [47]. As well Peyré-Tartaruga et al., 2022 found that NW and pole walking using predominantly elbows clearly have larger mediolateral margins of stability than walking, decreasing moderately the muscular engagement of gluteus medius [48].

              To date, according to the authors, no data have been found in the literature on changes in five point spinal kinematics during Nordic walking and comparison with regular walking.

The text was deleted:

Nordic walking (NW) has gained popularity under various names such as ski walking, pole walking, or fitness walking. Originally developed to fill the gap during the off-season for cross-country skiers, NW combines the techniques of cross-country skiing and regular walking. It can be practiced year-round and in a variety of environments. Nordic walking originated in Finland several decades ago and has since become a global phenomenon, with over 14 million people worldwide now using it as a means of maintaining physical fitness.

This manuscript lacks a focused rationale for spine-specific analysis. For example, why spine and not lower limb joints, which are more studied? Please include a more precise justification for examining spine kinematics specifically. Also, elaborate on why the chosen speeds (3 and 5 km/h) were selected

We thank the reviewer: The answer to the reviewer question is in the question – spine was choosen because  lower limbs and joints are more studied and it has lack for spine kinematic alterations under NW.

The text was added: Two slow walking speeds of 3 and 5 km/h were chosen due to the fact that Nordic walking is practiced unprofessionally mainly by the elderly. The second reason for choosing of this two NW velocities is the fact that they are probably the most used in NW.

Please add significance to the study after the aims, not the methods or statistical analyses. Also, suggest some hypotheses based on the previous research.

We thank the reviewer: The significance of the study was summarized in the Conclusions.

The methods and statistical analyses are important and for this reason, the authors have explained them well and clearly. The author team avoided formulating hypotheses because very little of the available research concerns the kinematic changes in the spine during Nordic walking and a number of other reasons.

The text was added: To date, according to the authors, no data have been found in the literature on changes in five point spinal kinematics during NW and comparison with regular walking.

Methods:

Sample size was not justified. Please provide an adequate power analysis for this sample.

There are no participant characteristics beyond age and gender: training history? NW experience? Health status?

We thank the reviewer. The power analysis for this sample with training history is not the purpose of our study. We thank the reviewer, in view of his correction, the text was added: Every participant in the target group used denied experience with Nordic walking and excellent health status. 

It remains unclear if randomisation or familiarisation was used for walking speed order.

We thank the reviewer, the walking speed order - first 3km/h and then 5 km/h.

The text was added: The experimental model used contains four types of walking for 60 seconds, which following one after the other in the following sequence and it was recorded of certain quantitative parameters:

- normal walking at a speed of 3 km/h on a training track for 1 minute;

- Nordic walking at a speed of 3 km/h on a training trail for 1 minute;

- normal walking at a speed of 5 km/h on a training track for 1 minute;

- Nordic walking at a speed of 5 km/h on a treadmill for 1 minute.

There is a lack of clarity on data processing: e.g., how was kinematic data filtered? What was the sampling rate?

We thank the reviewer, in view of his correction, the text was added: The obtained kinematic data was automatically filtered by 3D MioMotion software. The sampling rate was 200 Hz.

The obtained (reported) results, presented as SSA(t) plots and p-values, provide estimates of group differences. However, a closer examination of individual data records reveals unique patterns and differences that are not addressed in this study.

The statistical approach is under-described. Repeated-measures ANOVA is mentioned, but assumptions (normality, sphericity) are not addressed, and effect sizes are not reported. On the other hand, I find it unnecessary to add formulas for ANOVA and the T-test. Please rewrite this section to focus more on adequate analyses, post hoc, and reports.

We thank the reviewer. After reading this comment we understood that paragraph 2.2 is poorly explained and significantly rearrange the whole paragraph 2.2.  We also used fANOVA in the abstract, which is little bit misleading. Now, we put the focus on the used methods and replaced the general and bibliographic remarks and texts with more detailed explanations. This include extended explanation of used time series data, more precise definition of the used Null Hypothesis and clearer outline of used permutation repetitions to address low sample size.  

Results:

Figures and tables are dense and not reader-friendly. Results are only briefly described in text, relying on the reader to interpret complex tables and graphs. Please add some summary information in the text or condense the results.

We thank the reviewer. In our opinion, the results are very clearly summarized in Tables 1 and 2. А detailed summary information and condensed results are described in the Conclusion.

Discussion and Conclusion:

The discussion overgeneralizes from minimal or non-significant findings.

We thank the reviewer, in view of his correction, large part of the conclusions was moved into section Discussion and this section was significantly rearranged.

No discussion of why specific spine segments (e.g., thoracic vs. lumbar) respond differently. After the correction in the statistical analysis, in particular, adding the effect size and justifying the ANOVA in the first place, the discussion should be thoroughly checked and rewritten.

We thank the reviewer, in view of his correction, the text was added:

4.2. Hypothesis for different kinematic response of different tested segments of the spine under NW

The results obtained in this work raise more questions than they provide answers. The author team refrained from hypotheses due to the described limitations of the study, due to the specificity of the experiments performed with the correspondence of the small group we included in the research and due to the fact that it is very easy to make hypotheses but difficult to prove them. However, a more general hypothesis can be defined.

The different kinematic response under NW of different tested segments of the spine (e.g., thoracic vs. lumbar) is most likely due to:

- the different correlation of these segments to the upper limbs and lower limbs movement;

-  the height and weight of the person and the length of the poles, respectively;

- the adaptation to poles of the person performing NW;

- the speed of Nordic walking;

- the strictly individual gait of each person;

- the strictly individual arm moving (without and with poles) of each person.

Reviewer 2 Report

Comments and Suggestions for Authors

Spine kinematics is crucial in maintaining both functional mobility and general health. Alterations in spine kinematics could impact back pain, decreased mobility, and risk of falls. Nordic walking presents potential advantages for improving gait mechanics and spinal stability.

The authors presented an interesting study, and the findings suggest the rehabilitative potential of NW in gait and spinal stability.

The sample size is very small; could it be considered as a pilot study?

How can NW be useful in the rehabilitation of particular conditions such as older adults, neurological diseases, and subjects with a higher fall risk?

Thank you.

Author Response

Spine kinematics is crucial in maintaining both functional mobility and general health. Alterations in spine kinematics could impact back pain, decreased mobility, and risk of falls. Nordic walking presents potential advantages for improving gait mechanics and spinal stability.

We thank the reviewer, in view of his correction, the text was added: Spine kinematics is crucial in maintaining both functional mobility and general health. Alterations in spine kinematics could impact back pain, decreased mobility, and risk of falls. Nordic walking presents potential advantages for improving gait mechanics and spinal stability.

The authors presented an interesting study, and the findings suggest the rehabilitative potential of NW in gait and spinal stability. The sample size is very small; could it be considered as a pilot study?

We thank the reviewer, in view of his correction, the title was changed:

Spine kinematic alterations in Nordic Walking under two different speeds of 3 and 5 km/h - a pilot study

How can NW be useful in the rehabilitation of particular conditions such as older adults, neurological diseases, and subjects with a higher fall risk?

We thank the reviewer, in view of his correction, the text was added in the conclusion:

How can NW be useful in the rehabilitation of particular conditions such as older adults, neurological diseases, and subjects with a higher fall risk? NW will certainly have a positive effect on pathological conditions of the spine, neurological diseases, but not in the acute phase and tailored to the individual characteristics of the person. Its appointment by clinicists and its dosage should be carried out taking into account the respective pathology and its degree, patient feedback and the degree of the achieved result.

Reviewer 3 Report

Comments and Suggestions for Authors

I reviewed the article “Spine kinematic alterations in Nordic Walking under two different speeds of 3 and 5 km/h”. This experimental study investigates how spinal kinematics are influenced by Nordic walking (NW) versus regular walking (RW) over a short time frame (60 seconds) in healthy, physically fit young adult males (n=20, aged 19–22). It aims to quantify angular changes in specific spinal segments.

Abstract

The abstract follows the appropriate structure; however, several issues regarding grammar, style, or typographical errors have been identified:

  • "detetcted" is a typographical error (line 18).
  • The phrase “the data is acquired” (line 16) should be revised to “were acquired.”
  • Similarly, “are detected” (line 18) is better expressed as “were detected.”
  • The construction “by revealed significant differences” (line 21) would be improved by rephrasing it as “as it revealed significant differences.”
  • Change “The influence… was evaluated” (line 23) to “Nordic walking increased stance phase duration and velocity.”

The abstract methods lack details. The results are too brief, omitting statistical significance values (p-values), effect sizes, and information about trial conditions (e.g., whether NW was tested first or last). Move relevant conclusions to clarify results and keep the conclusion short.

Introduction

The first, second and third paragraphs (33–42 and 43–49 and 50-55)) are similar in structure, with differences primarily in wording and topics. A revision could address these similarities.

Materials and Methods

This section describes the subjects' characteristics, the experimental model, the parameters used, and the statistical methods applied.

Some phrases could be revised. For instance: the phrase “Using the 3D MioMotion IMU sensor’s analysis system (Noraxon) motion analysis is carried out using compact inertial sensors, which are placed on fig. 2 selected placements under body segments” can be rewritten as: “Motion analysis is conducted with the 3D MioMotion IMU sensor analysis system (Noraxon), utilising compact inertial sensors that are positioned on specific body segments as indicated in Figure 2.” (lines 151-152).

Results

The results section clearly presents key data and findings.

Certain phrases could be revised for clarity. For instance, “The used and described experimental setup was designed to assess the angular rotations of the spine in the three directions” (line 229) may be rephrased as “The experimental setup employed in this study was specifically designed to evaluate spinal angular rotations across all three planes.”

Discussions summarize the study's findings on changes in walking kinematics in healthy individuals.

The discussion should further examine the study's implications and how its findings can be incorporated into current rehabilitation protocols to inform clinicians and researchers.

The authors highlight the paper's limitations: a small sample of young, fit males and reliance on treadmill testing. Additional limitations should also be considered, such as the brief duration of the experiment, which may not adequately capture long-term biomechanical adaptations, and the lack of data on upper limb mechanics, an essential component of Nordic Walking.

The bibliography is comprehensive, supporting the study’s claims with scientific evidence.

Author Response

I reviewed the article “Spine kinematic alterations in Nordic Walking under two different speeds of 3 and 5 km/h”. This experimental study investigates how spinal kinematics are influenced by Nordic walking (NW) versus regular walking (RW) over a short time frame (60 seconds) in healthy, physically fit young adult males (n=20, aged 19–22). It aims to quantify angular changes in specific spinal segments.

Abstract

The abstract follows the appropriate structure; however, several issues regarding grammar, style, or typographical errors have been identified:

We thank the reviewer, in view of his correction, all proposals was added:

  • "detetcted" is a typographical error (line 18).
  • The phrase “the data is acquired” (line 16) should be revised to “were acquired.”
  • Similarly, “are detected” (line 18) is better expressed as “were detected.”
  • The construction “by revealed significant differences” (line 21) would be improved by rephrasing it as “as it revealed significant differences.”
  • Change “The influence… was evaluated” (line 23) to “Nordic walking increased stance phase duration and velocity.”

The abstract methods lack details. The results are too brief, omitting statistical significance values (p-values), effect sizes, and information about trial conditions (e.g., whether NW was tested first or last). Move relevant conclusions to clarify results and keep the conclusion short.

We thank the reviewer, in view of his correction, text was added:

“in the following sequence”.

From line 155 to line 164 in the revised manuscript, the sequence of tests is described.

Large part of the Conclusions was moved into section Discussion.

Introduction

The first, second and third paragraphs (33–42 and 43–49 and 50-55)) are similar in structure, with differences primarily in wording and topics. A revision could address these similarities.

We thank the reviewer, in view of his correction, the text was removed:

Nordic walking (NW) has gained popularity under various names such as ski walking, pole walking, or fitness walking. Originally developed to fill the gap during the off-season for cross-country skiers, NW combines the techniques of cross-country skiing and regular walking. It can be practiced year-round and in a variety of environments. Nordic walking originated in Finland several decades ago and has since become a global phenomenon, with over 14 million people worldwide now using it as a means of maintaining physical fitness.

Materials and Methods

This section describes the subjects' characteristics, the experimental model, the parameters used, and the statistical methods applied.

Some phrases could be revised. For instance: the phrase “Using the 3D MioMotion IMU sensor’s analysis system (Noraxon) motion analysis is carried out using compact inertial sensors, which are placed on fig. 2 selected placements under body segments” can be rewritten as: “Motion analysis is conducted with the 3D MioMotion IMU sensor analysis system (Noraxon), utilising compact inertial sensors that are positioned on specific body segments as indicated in Figure 2.” (lines 151-152).

We thank the reviewer, in view of his correction, the text was corrected: “Motion analysis is conducted with the 3D MioMotion IMU sensor analysis system (Noraxon), utilising compact inertial sensors that are positioned on specific body segments as indicated in Figure 2.”

Results

The results section clearly presents key data and findings.

Certain phrases could be revised for clarity. For instance, “The used and described experimental setup was designed to assess the angular rotations of the spine in the three directions” (line 229) may be rephrased as “The experimental setup employed in this study was specifically designed to evaluate spinal angular rotations across all three planes.”

We thank the reviewer, in view of his correction, the text was corrected: “The experimental setup employed in this study was specifically designed to evaluate spinal angular rotations across all three planes.”

Discussions summarize the study's findings on changes in walking kinematics in healthy individuals.

The discussion should further examine the study's implications and how its findings can be incorporated into current rehabilitation protocols to inform clinicians and researchers.

We thank the reviewer, in view of his correction, the text was added into Conclusions:

How can NW be useful in the rehabilitation of particular conditions such as older adults, neurological diseases, and subjects with a higher fall risk? NW will certainly have a positive effect on pathological conditions of the spine, neurological diseases, but not in the acute phase and tailored to the individual characteristics of the person. Its appointment by clinicists and its dosage should be carried out taking into account the respective pathology and its degree, patient feedback and the degree of the achieved result.

The authors highlight the paper's limitations: a small sample of young, fit males and reliance on treadmill testing. Additional limitations should also be considered, such as the brief duration of the experiment, which may not adequately capture long-term biomechanical adaptations, and the lack of data on upper limb mechanics, an essential component of Nordic Walking.

We thank the reviewer, in view of his correction, the text was added into Limitations of the study: The third is the brief duration of the experiment, which may not adequately capture long-term biomechanical adaptations. The last is the lack of data on upper limb mechanics, an essential component of Nordic Walking.

Round 2

Reviewer 1 Report

Comments and Suggestions for Authors

I appreciate some of the authors' efforts to improve the manuscript; however, some issues remain:

  1. When I mentioned the power analysis, I was referring to using some tool (such as GPower) to justify the sample size.
  2. Line 168, there is some unrelated text left.
  3. Unnecessary formulas for ANOVA are still part of the text, making it hard to read. 
  4. "Figures and tables are dense and not reader-friendly". The authors did not address this issue. 

Overall, I find the lack of effort from authors to really improve the manuscript. Please address all the comments from this and previous reviews meticulously and scientifically.

Author Response

We are very thankful for profound review of our work. Apparently, this improved the quality of our report. We did a second revision with major changes of our reporting part with a focus on methodology. Having the Reviewer’s comments we understand that unclear parts remained in the second version. To address the issue we did the following changes:

  1. When I mentioned the power analysis, I was referring to using some tool (such as GPower) to justify the sample size.

Answer 1. We thank the reviewer. According the Effect Size and Power analysis we addressed the issue with more precise description of methodology. Usually, effect size and Power analysis are connected to scalar type of data. As we understood from the manual of GPower tool, as well as the pwr library in R, the libraries are not applicable to functional data. Not because it is not possible to compute the required values, but because such kind of measure is misleading for functional data time series.

              However, we confess the lack of any reference value for description of the used data. Thus, we computed the L2-norm values to estimate scalar estimates of the differences between the mean value functions. Because, the best approach to do this analysis of FDA data is the visual presentation, we reported the computed measures together with redesigned plots for pointwise mean values.

              According the power analysis, we understand that the test of 20 samples may not be good enough. But, this is a common issue in this kind of analysis and explains why we opted to use rmfanova library. Its implementation is based on permutation/bootstrapping to use the asymptotic non-parametric distribution. Thus, the reported p-values are not standalone estimate, as might be wrongly assumed, but indicator results obtained after 1000 repetitions. Having relatively clear values, we do not assume that increasing the number of permutations would change the outcome significantly.   

  1. Line 168, there is some unrelated text left.

We thank the reviewer, in view of his correction, the text in line 168 was corrected. We are really sorry for this typos.

  1. Unnecessary formulas for ANOVA are still part of the text, making it hard to read. 

Answer 3. We thank the reviewer, in view of his correction we revised the 2.2 section removing some review text linked to fANOVA together with related formulas. However, due to revision related to the point 1 we introduced some extra text with two additional formulas in aim to clarify our results.

  1. "Figures and tables are dense and not reader-friendly". The authors did not address this issue. 

Answer 4. We thank the reviewer, in view of his correction, the Figures presenting the data is completely redesigned. We excluded the individual trajectories and left only mean values curves supported with additional scalar results for L2-norm values. In addition, we extended and changed x- and y- axis text and its size of all figures for more clear presentation. 

Overall, I find the lack of effort from authors to really improve the manuscript.

The authors disagree with this conclusion. In the first series of responses from reviewer 1, 12 meticulously and scientifically corrections were made in response to reviewer 1's recommendations and another 15 corrections were made by the other two reviewers, who rated the manuscript excellent. In the second series of responses from reviewer 1, five meticulously and scientifically corrections (including 30 figures redrawing) were made.

As a result of the all reviewers' appropriate recommendations, the article gained further quality. We sincerely thank reviewer 1 for his hard work and efforts.

Round 3

Reviewer 1 Report

Comments and Suggestions for Authors

Dear authors,

I appreciate the additional effort you put into improving this manuscript. Although I don’t entirely agree with all the choices and methodological approaches you made, I believe this manuscript is good enough that it can be published in its current form.

Best regards